# The Influence of Environmental Features on the Morphometric Variation in *Mauritia flexuosa* L.f. Fruits and Seeds

**DOI:** 10.3390/plants9101304

**Published:** 2020-10-02

**Authors:** Nilo L. Sander, Carolina J. da Silva, Aline V. M. Duarte, Bruno W. Zago, Carla Galbiati, Iris G. Viana, Joari C. de Arruda, Juliana E. Dardengo, Juliana P. Poletine, Marcelo H. Siqueira Leite, Marta H. S. de Souza, Robson F. de Oliveira, Thallita S. Guimarães, Valvenarg P. da Silva, Marco A. A. Barelli

**Affiliations:** 1Graduate Program in Amazon Biodiversity and Biotechnology, Bionorth Network—Av. Santos Dumond–Cidade Universitária, Cáceres, Mato Grosso State 78200-000, Brazil; ecopanta@terra.com.br (C.J.d.S.); melao.a.v.bio@gmail.com (A.V.M.D.); brunowzago@hotmail.com (B.W.Z.); iris.vianabio@gmail.com (I.G.V.); arrudajcbio@gmail.com (J.C.d.A.); juliana.dardengo@unemat.br (J.E.D.); sleitebrum@gmail.com (M.H.S.L.); martaufmt@gmail.com (M.H.S.d.S.); robson.oliveira04@gmail.com (R.F.d.O.); silvabiologo@hotmail.com (V.P.d.S.); mbarelli@unemat.br (M.A.A.B.); 2Graduate Program in Environmental Science; UNEMAT, Cáceres, Mato Grosso State 78200-000, Brazil; carla@unemat.br; thallita_guimaraes@hotmail.com (T.S.G.); 3Graduate Program in Plant Genetics and Breeding; UNEMAT, Cáceres, Mato Grosso State 78200-000, Brazil; jppoletine@uem.br (J.P.P.)

**Keywords:** Amazon, biometry, buriti, Brazilian Savannah, ecological tension zone

## Abstract

The environmental heterogeneity may reflect the different morphological and phenotypic traits of individuals belonging to a single species. We used 14 morphological traits of *Mauritia flexuosa* L.f. to understanding the relation between environment and phenotypic traits. Twenty-five fruits were collected from each of the 10 individuals sampled in each study site: Chapada dos Guimarães (CG), Vila Bela da Santíssima Trindade (VB), and Alta Floresta (AF). We analyzed the genetic divergence, using the standardized Euclidean distance, the sequential method of Tocher, unweighted pair group method with arithmetic mean (UPGMA), and the projection of the distances onto 2D plane, and calculated the relative importance of the traits evaluated. The analysis showed the partition of individuals into three main groups: Two groups comprising the majority of individuals. Fresh fruit weight, pulp rate, fresh pulp weight, and moisture rate were the traits that most helped explaining the difference between materials. The results shown in the current study evidenced the influence of these three different environments on the biometric traits of *M. flexuosa*. Such influence has led to the formation of Alta Floresta and Vila Bela da Santíssima Trindade individuals in different groups, whereas the Chapada dos Guimarães individuals were able to permeate the two other groups, although they showed stronger tendency to group with individuals from Vila Bela da Santíssima Trindade.

## 1. Introduction

Mato Grosso State presents wide environmental heterogeneity due to the rich and complex plant biodiversity found in its three different biomes, namely, Pantanal, Brazilian Savannah; the Amazon; and the ecological tension zones (ETZ), which is the transition site between the two largest Brazilian plant biomes, the Brazilian Savanna and the Amazonian Forest [1]. This environmental heterogeneity favors the formation of vegetation mosaics, which are controlled by climate, soil properties, fire regimes, groundwater depth, and topography [2]. The different phytophysiognomies found in the state constitute an ecological gradient that can be seen through the genetic variability of plants found in it. The buriti species *Mauritia flexuosa* L.f.is distributed in these three different areas.

Buriti is a dioecious palm tree, up to 40 m tall [3], which is considered by some local and indigenous communities as the “tree of life”. The species is widely distributed in the Amazon region, in the Brazilian Savannah, and in Pantanal. It grows in the Brazilian northern and midwestern regions and in parts of the northeastern and southeastern regions of the country, as well as in other South American countries, in low-lying areas of open and closed forests and in poorly drained, swampy, or flooded soils [4].

All parts of this palm tree can be used. Its leaves can be used in craftwork and its fruits are consumed fresh or processed as juice, porridge, soup, and ice cream [5]. The fruit production may vary from region to region and it may reach more than 2000 annual fruits/plant [3]. The harvest and sale of the fruit may compose more than 60% of the family income during the harvest periods [6]. However, despite the plants’ cultural and environmental importance, the areas where this species grows have been devastated year after year, a fact that significantly reduces the species’ population [7].

The phenotypic differences are set by environmental variations because even fruits belonging to a single species and distributed in different locations are subject to temperature, photoperiod, and rainfall rate variations, as well as to other variables that may be linked to certain aspects of the species’ genetic composition [8]. The different environmental biotic and abiotic conditions may dramatically affect buriti’s phenotypic traits and lead to consequences of seed germination, plant size, and fruit production [9].

Morphometry studies allow understanding the different phenotypic traits of buriti fruits and seeds. Such studies are an important instrument for the detection of genetic variability within populations, as well as to find the relation between this variability and the environmental factors. They also provide important information that help featuring the ecological and genetic aspects concerning conservation and breeding programs. The aim of the current study was to analyze the influence of the environment on the morphometry of *Mauritia flexuosa* L. f. fruits and seeds grown in the Amazon and Brazilian Savannah biomes, as well as in the ecological tension zones.

## 2. Results

The quantitative data about the 14 morphometric traits of fruits and seeds were collected from buriti individuals. Individuals 1–10 represent Alta Floresta; 11–20, Vila Bela da Santíssima Trindade; and 21–30, Chapada dos Guimarães. The combination between individual 18 (VB) and 29 (CG) was the most dissimilar one, since it presented the highest standardized Euclidean distance estimate (dii’ 6.36); whereas the lowest divergence was observed between the pair 24 and 30, both collected in CG (dii’ 0.00).

The analysis resulted in the formation of five different groups (Table 1. Group I presented the largest number of representatives (11), since it comprised 36.67% of the 30 analyzed individuals. Groups II and III comprised nine and five individuals, respectively. Group IV was formed by four individuals, whereas group V only held one individual. 

The dendrogram of the individuals generated through the UPGMA (Unweighted Pair Group Method with Arithmetic Mean) clustering method according to the standardized Euclidean distance is shown in Figure 1 The 70% dissimilarity cut indicated the formation of three main groups, and the 40% dissimilarity cut showed the formation of six subgroups from the main three ones.

Subgroups 1a, 1b, and 2a comprised seven individuals each, and these individuals derived from different locations. Subgroup 1a comprised individuals from CG and VB. Subgroup 1b comprised individuals from VB, whereas subgroup 2a comprised individuals from AF. Subgroup 2b comprised five individuals, three from AF and two from CG. On the other hand, group 3 was represented by four (CG) individuals; three of them were allocated in subgroup 3a, whereas individual 26 was allocated alone in subgroup 3b.

By analyzing the projection of the distances in the plane (Figure 2), it was possible to see the partition of individuals in three main groups. Two of them encompassed most of the individuals and a separate group was formed by three individuals (22, 23, and 29). The groups were also formed according to the UPGMA and Tocher methods. According to the Tocher sequential optimization method, the UPGMA hierarchical method, and the projection of distances in 2D, the most divergent pair of individuals (VB-18 and CG-29) was allocated in different groups.

The quantitative morphometric traits explaining the divergence among the 30 buriti individuals were Fresh fruit weight (37.50%), Pulp rate (17.86%), Fresh pulp weight (14.68%), and Moisture rate (11.91%) (Table 2). The traits “fruit height” and “fruit diameter” were the least likely to explain the divergence; they contributed 0.0524% and 0.025%, respectively (Table 2). 

The scatter plot (Figure 3) was based on variables such as “FFW” and “pulp rate”, since they expressed greater relative contribution to trait importance. It is possible to see the dispersion of buriti individuals and the AF individuals can be seen within the interval from 1 to 10; they are spatially close to each other. Similarly, the VB individuals between the intervals from 11 to 20 showed the same trend. On the other hand, the CG individuals between the intervals from 21 to 30 were scattered among the other individuals; their presence was more concentrated at the top of the scatter plot.

## 3. Discussion

The genetic structure of natural populations can be influenced by many ecological factors. Groups of individuals established in different regions holding specific environmental features tend to genetically differentiate themselves as populations. It happens due to the gene flow limitation and to the different selection pressures faced by each population [8].

The dissimilarity matrix generated through the standardized Euclidean distance showed strong correlation between the individual’s genetic distance and place of occurrence. The 10 most divergent pairs of genotypes came from distant populations and, when the first 20 pairs were taken into consideration, just one pair of individuals was in the same location. Similar results were found by Santos [11], who studied genetic variation in natural babassu (*Attalea speciosa* Mart.) populations and found strong correlation between the individual’s genetic distance and place of occurrence. The first 15 most divergent pairs derived from different locations. The CG site had the highest intra-population variation; this variation can be attributed to genetic differences among individual parent plants, even within a small geographic area [12]. Matos et al. [13] found intra-population biometric differences in *M. flexuosa* fruits and seeds.

The clustering methods used in the present study corroborate the aforementioned observation, as it demonstrated the tendency to group individuals coming from the same location. The VB and AF individuals were found in isolated groups, whereas the CG individuals were merged between the two locations, a fact that can be explained by the individual features of each area wherein the samples were collected. Some of these features are: Geographical isolation, different climate, phytophysiognomy, and matrix where these populations were found. The environmental conditions may influence the morphological and physiological traits of fruits and seeds [2,6,8,14,15,16,17]. 

The AF and CG individuals came from areas called palm swamps; however, the AF palm swamp has perennial stream, whereas the CG has intermittent streams. On the other hand, the VB individuals came from a floodable forest called “buritizal”, which is located in the riverbanks of the Guaporé River. This river remains flooded for 6–8 months throughout the year and, in the other months, it keeps water availability because of the shallow water tables [18].

Although the region where the fruits were collected in VB is an ecological tension zone (Amazon-Brazilian Savannah), its phytophysiognomy shows predominant features of Savannah regions. Mariotti [7] performed a cover crop analysis in the county and pointed out that more than 65% of it is formed by Brazilian Savannah. That study site was included in this classification.

Based on this information, it was possible to see that most CG individuals were closer to the VB one, because of the phytophysiognomy similarities between these sites. However, the fact that some CG individuals were grouped with AF individuals may be linked to the availability of similar resources in these two areas.

The scatter plot based on the two traits showing the greatest relative dissimilarity contribution allowed seeing that AF individuals presented mean FFW higher than that found in the other two locations, a fact that can be explained by the different soil classes found in the research sites. The presence of acrisols in AF may have favored the FFW because such soils present sandy textures in surface soil horizons, as well as more clayey textures in subsurface horizons, where most buriti roots are often found. Thus, they enable water storage and cation exchange capacities higher than those enabled by sandy soils in the entire soil profile, as it was found in the CG and VB soils. On the other hand, the CG individuals, overall, showed pulp rate higher than that of the other individuals because this area is located at higher altitudes; thus, it favors conditions conducive to higher annual fruit productivity.

Fresh fruit weight, pulp rate, fresh pulp weight, and moisture rate were the traits that most helped to explain the difference between materials. Partially similar results were found in research carried out by Leão et al. [19], in the state of Pará Brazil, for the mogno tree species (*Swietenia macrophylla* King.) in which the fresh fruit mass was the second characteristic, with 21.47% in explaining the genetic diversity. In addition, the characteristics cited above were linked to the species adaptability to the spatiotemporal variability of habitats [20].

## 4. Materials and Methods

### 4.1. Study Site

The fruits were collected in buriti (*Mauritia flexuosa* L.f.) monodominant formations located in Alta Floresta (AF), Chapada dos Guimarães (CG), and Vila Bela da Santíssima Trindade (VB) counties, Mato Grosso State, Brazil (Figure 4).

Alta Floresta is located in the northern end of Mato Grosso State, within the Amazon biome. The tropical open ombrophilous forest is its dominant vegetation type, which is associated with palm trees and lianas. This forest type is featured by well-spaced large trees and frequent palm-tree groupings, as well as by huge amounts of sarmentous phanerophytes [21]. According to the soil map developed by Mato Grosso State Planning Department, acrisol is the soil class found in the AF study site. This soil type presents good drainage, low base saturation, low clay activity, medium texture, and gently undulating relief [22].

The climate in Chapada dos Guimarães and Vila Bela da Santíssima Trindade, according to the classification of Köppen (1948), is of the Aw type—tropical climate, with dry winter. It has a rainy season in summer, from November to April, and a clear dry season in winter, from May to October (July is the driest month). The average temperature of the coldest month is over 18 °C. Precipitation is greater than 750 mm per year, reaching 1800 mm. In Alta Floresta, the climate is of the tropical humid or subhumid (Am) type and presents four months of drought. It is also characterized by presenting an average temperature of the coldest month always above 18 °C, presenting a short dry season that is compensated by the high total precipitation (Table 3).

According to Ribeiro and Walter [23], the local landscape in Chapada dos Guimarães comprises a wide range of vegetation types. Different Brazilian Savannah (strict sense) phytophysiognomies can be seen in the region, namely, gallery forests, deciduous and semideciduous dry forests, and rupestrian fields [24]. The prevalent soil class in the CG study site is the Quartzenic *Neosol,* which presents sandy or sandy loam texture. This soil class comprises nonhydromorphic, discolored quartz sands and presents yellow or red color [22].

Vila Bela da Santíssima Trindade in Southwestern Mato Grosso State, Alto Rio Guaporé Valley, holds transitional features between two of the largest Brazilian biomes, namely, Brazilian Savannah and Amazon Forest [25]. In addition, its landscape alternates from mountains to plains, and it features the collection site (VB) as ETZ. According to the soil map developed by Seplan-MT [22], Gleysol is the predominant soil class in this study site. This hydromorphic soil is permanently or periodically saturated by water and presents low clay activity and flat relief.

### 4.2. Data Collection and Samples

Twenty-five fruits were collected from each of the 10 different buriti palm trees in each study site. The collected samples were at least 100 m away from each other. This condition was necessary to decrease the number of specimens derived from common parents because parental diversity increases genetic variability within the sampling groups [26]. The individuals were chosen according to Barbosa et al. [15], by relating the maturation of the infructescence according to the fruits fallen on the soil, as well as to their shape and color contrast. The criteria were visually assessed by a single observer, in order to avoid different assumptions.

After the fruits were assessed, they were transferred to identified and sealed polyethylene bags and, subsequently, stored in a cooler until the end of the collection process. Next, they were kept under refrigeration in the laboratory of the Study Center for Limnology, Biodiversity, and Ethnobiology of Pantanal (CELBE - Pantanal), Mato Grosso State University (UNEMAT) – Cáceres campus.

The height (longitudinal direction) and diameter of each sampled fruit were measured in a digital caliper (Hardened model, Stainless, Fairfield, NJ, USA), whereas the total weight was measured using an analytical scale (AUY220 model, Shimadzu, São Paulo, Brazil). The fruits were peeled and pulped after their morphometric parameters were taken. Then, the peel, pulp, seed, and other structures were separated. All the structures mentioned above were separately weighed and classified as fresh weight. Moisture was determined through the oven method, at 105 ± 3 °C for 24 h, according to the recommendations in the Rules for Seed Analysis [16]. Next, the weighing was once more conducted in order to find the dry weight.

The means of each assessed trait were obtained, namely, fresh fruit weight (FFW), fresh peel weight (FPW), fresh pulp weight (FPuW), fresh seed weight (FSW), other fresh structures (OFS), height (H), diameter (D), dry fruit weight (DFW), dry peel weight (DPW), dry pulp weight (DPuW), dry seed weight (DSW), other dry structures (ODS), and moisture content (MC).

### 4.3. Statistical Analysis

The multivariate analysis was used to estimate the genetic divergence through the *standardized Euclidean distance* as dissimilarity measure [27] and to estimate the influence of environmental conditions on the morphometry of *Mauritia flexuosa* L. f. fruits and seeds from plants distributed in the Amazon and Brazilian Savannah biomes, as well in the ecological tension zones (ETZ):$$d_{\mathrm{ii}’}=\;\sqrt{\frac{1\;\;}{n}\underset{j}{\sum }(x_{\mathrm{ij}}{-\;x}_{i^{’}j})}2$$
where dii’ is the distance between genotypes i and i’, Xij is the observation of the ith parent referring to the jth trait, and *n* is the number of studied traits.

The grouping methods were intended to separate an original group of observations into several subgroups, in order to obtain homogeneity within and heterogeneity among the subgroups. Among these methods, optimization and hierarchical methods were employed on a large scale (BERTAN et al. 2006). Subsequently, to obtain the dissimilarity matrix, we used methodologies that made it possible to visualize the separation of the individuals evaluated in groups of similarity, which promoted a clearer and more visual interpretation of the results.

The sequential Tocher method, the hierarchical clustering method (UPGMA), and the projection of distances in 2D format were used as clustering techniques according to the standardized Euclidean distance [27]. The clustering analysis applied to the individuals was conducted according to the Tocher sequential optimization method. Furthermore, the relative importance of the assessed traits was calculated according to the methodology proposed by Singh [10]. All the analyses were performed using the computational resources in the Genes Software [28].

## 5. Conclusions

There was environmental influence on the morphometry of fruits and seeds from *Mauritia flexuosa* L. f. plants distributed in the Amazon (AF) and Brazilian Savannah (CG) biomes, as well as in the ecological tension zone (VB). Two large, distinct groups were formed: Alta Floresta (AF) and Vila Bela da Santíssima Trindade (VB) individuals, whereas individuals in Chapada dos Guimarães (CG) permeated these two groups, although they showed greater trend to group with VB individuals. The greatest intrapopulation variation was observed between CG individuals, whereas the AF group showed the smallest variation. The fresh fruit weight and the pulp rate were the main traits contributing to the clustering results.

## Figures and Tables

**Figure 1 plants-09-01304-f001:**
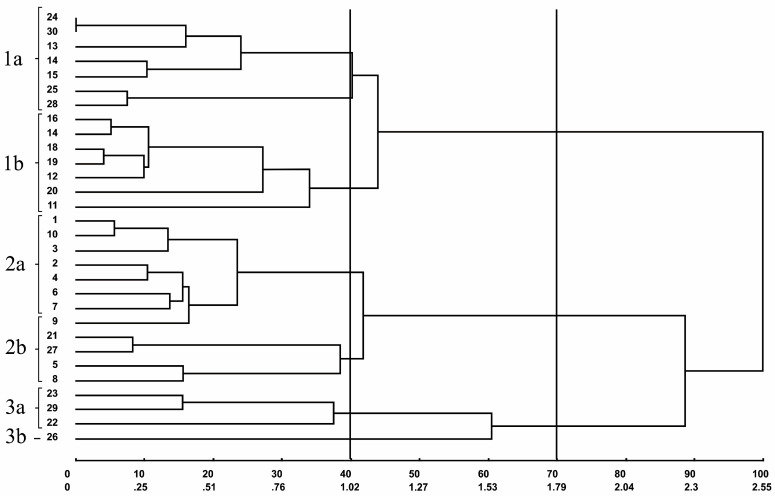
Dendrogram representing the dissimilarity pattern among the 30 buriti (*M. flexuosa*) individuals obtained through the unweighted pair group method, the arithmetic mean (UPGMA) was based on the standardized Euclidean distance estimated from 14 quantitative traits of the buriti fruits and seeds.

**Figure 2 plants-09-01304-f002:**
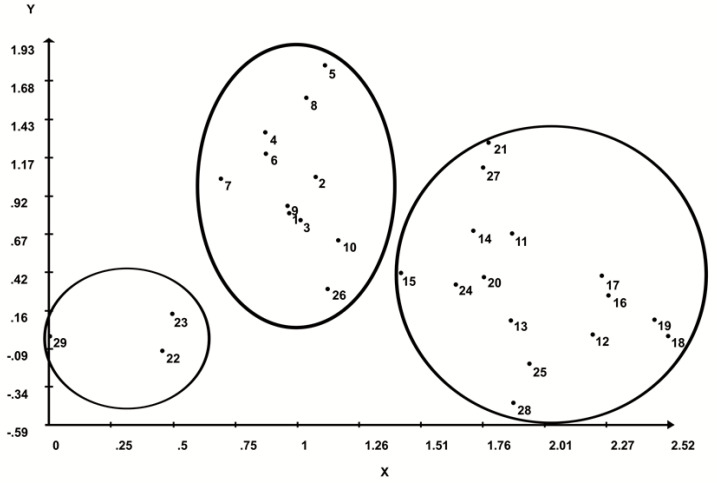
Projection of distances in 2D by taking into consideration the 30 buriti (*M. flexuosa*) individuals, based on the standardized Euclidean distance estimated from 14 quantitative traits of buriti fruits and seeds.

**Figure 3 plants-09-01304-f003:**
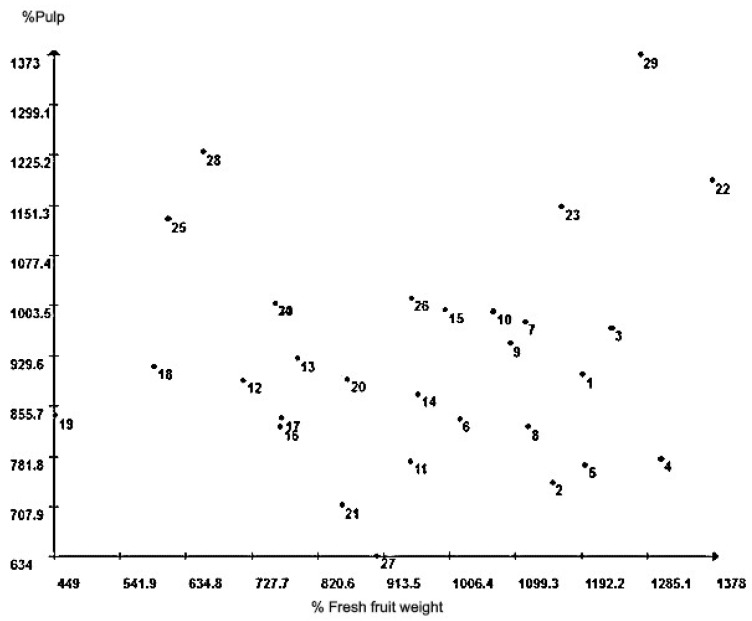
Scatter plot of 30 buriti (*M. flexuosa*) individuals based on the two traits that showed the greatest relative dissimilarity (S.j′) contribution, analyzed according to Singh [10] criteria.

**Figure 4 plants-09-01304-f004:**
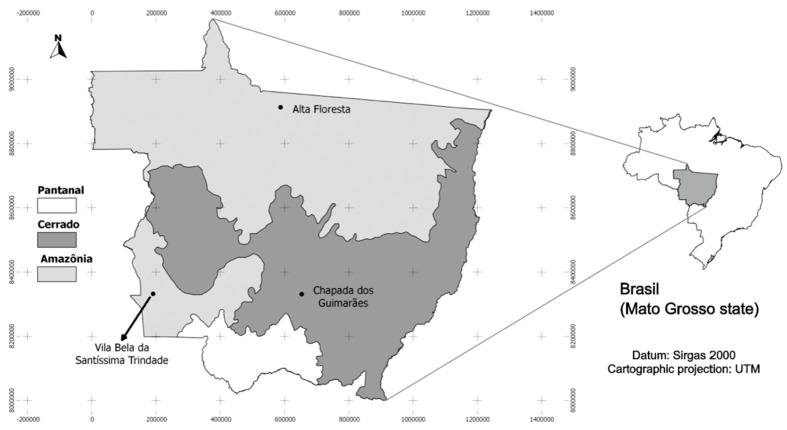
Biomes in Mato Grosso State and the locations of the study sites in the respective counties. (Instituto Brasileiro de Geografia e Estatıstica - IBGE).

**Table 1 plants-09-01304-t001:** Clustering of 30 buriti individuals according to the Tocher method, based on the standardized Euclidean distance estimated from 14 quantitative traits.

Group	Individuals	(%)
I	24, 30, 13, 25, 28, 12, 19, 16, 18, 17, 20	36.67
II	01, 10, 03, 02, 09, 04, 07, 06, 08	30.00
III	21, 27, 14, 15, 11	16.67
IV	23, 29, 22, 26	13.33
V	05	3.33
Total		100.00

**Table 2 plants-09-01304-t002:** Relative dissimilarity (S.j’) contribution rate of the 14 quantitative traits analyzed in the fruits and seeds.

Trait	S.j’	Contribution (%)
Fresh fruit weight (FFW) (g)	12,1054.08	37.50
Pulp rate (PR)	57,663.50	17.86
Fresh pulp weight (FPuW) (g)	47,393.07	14.68
Moisture rate (MC)	38,449.07	11.91
Fresh seed weight (FSW) (g)	20,033.90	6.20
Dry fruit weight (DFW) (g)	17,810.58	5.51
Dry seed weight (DSW) (g)	6798.02	2.10
Fresh peel weight (FPW) (g)	4802.09	1.48
Other fresh structures weight (OFS) (g)	4708.57	1.45
Dry peel weight (DPW) (g)	1476.64	0.45
Dry pulp weight (DPuW) (g)	1248.68	0.38
Other dry structures weight (ODS) (g)	1078.60	0.33
Fruit height (H) (mm)	169.10	0.05
Fruit diameter (D) (mm)	80.78	0.02
Total		100.00

Data from 30 buriti (*M. flexuosa*) individuals, analyzed according to Singh [10].

**Table 3 plants-09-01304-t003:** Climatic indices of *Mauritia flexuosa* L.f. sites collected.

Locality	Coord. X; Y	Rainfall (mm).	No. Dry Months	Altitude (m)	Temp. (min)	Temp. (max)
Chapada dos Guimarães	15°23’1.05” S; 55°49’46.8” O	1.300–1.400	6	600–800	18	30
Vila Bela da Santíssima Trindade	15°3’37.3” S; 59°54’18.5” O	2.150	3–4	250	17	33
Alta Floresta	9°47’15.56” S; 56° 4’1.73” O	2.000–2.300	4	200–300	18	32

Source: SEPLAN/Mato Grosso/JULY/AUGUST/2000.

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
