# Peer review of "The Influence of Environmental Features on the Morphometric Variation in Mauritia flexuosa L.f. Fruits and Seeds"

_plants, 2020, doi:10.3390/plants9101304_

Round 1

Reviewer 1 Report

Authors studied environmental and phenotypic effects in on the morphometry of fruits and seeds from Mauritia flexuosa L. f. plants distributed in the Amazon (AF) and Brazilian Savannah (CG) biomes, as well as in the ecological tension zone (VB).

We analyzed the genetic divergence, using the 18 Standardized Euclidean Distance, the sequential method of Tocher, UPGMA, the projection of the distances onto 2D plane and calculated the relative importance of the traits evaluated. The analysis has shown the partition of individuals in three main groups: two groups comprising the majority of individuals. Authors has worked in novel species but this manuscript need thoroughly look before acceptance for publications. Quality of figures are not acceptable, many english and grammatical errors throughout the ms and insufficient explanation of results should overlook.

Specific comments

15-17; method section is not clearly stated

65; what does ecological tension zones (ETZ) means and why it came here?.

69; was or were?

72-74; move this sentence to the method sections

Figure 2,3 and 4; axis numbers are not readable -enlarged them

125; spacing between the

135-140; reorganize it

254:?

Check all the references thoroughly there are many errors

Author Response

we make all the changs, but we will send only one archive in the last reviwer.

Thanks

Reviewer 2 Report

The influence of environmental features on the 3 morphometric variation in Mauritia flexuosa L.f. 4 fruits and seeds

General comments

References format within the text does not follow the Journal format.

Define abbreviations.

More details in Materials and Methods

Discussion need to be improved to better explain the results.

Have you studied all environmental factors or only Edaphic factors?

Write full name of all abbreviations in abstract part line: 17, 19, 25, 25, 27

Language: Line 46: Everything in this palm tree is useful…revise!!

Line: 48  200,000 annual fruits/plant. Revise  (do you mean 200,000 per year???

Line: 140, 149, 154, 190, 226 format text as journal format

Line 120: supporting discussion with recent and relevant references

Line 125: consider editing.

Line 170: add the location of study area with latitude and longitude

Line 254: complete author contributions

Line 262: scientific name must be italic

Line 275: scientific name must be italic

Line 288: journal name (first letter must be Caps)

Line 292: journal name (first letter must be Caps)

Line 305: scientific name must be italic

Revise reference list. Some references either not listed or listed and not mentioned in text.

For example:

- References: references below not founded

 - Ratter 1992 (Line 34)

- Rull and Montoya 2014 (line 45)

- Virapongse et al. 2017 (line 47)

- Baskin and Baskin 2014 (line 56, 124)

- Qaderi and Cavers 2002

-Matos et al. 2014 (line 133)

- Da Silva et al. 2015 (line 147)

- Cruz and Regazzi 2014 instead of Cruz et al. 2014 (Line 226)

- Reference # 29 not mentioned in text.

Author Response

(The authors gave the same response as above.)

Reviewer 3 Report

Manuscript presented for review with title: “The influence of environmental features on the morphometric variation in Mauritia flexuosa L.f. fruits and seeds”  is really interesting. The experiment was planned very carefully. The Introduction section includes  all nescesary information about examined species.

I have only one question to this part of manuscript: is really need to use a little bit old references as: (Ratter 1992), (Delgado et al. 2007), or (Luzuriaga et al. 2006)? In my opinion not older than 10-years old should be used.

The collected experimental material and used methods do not raise any objections.

In my opinion the obtained results should be presented in tables as measured results with their units. If authors examined for example fresh fruit rate (g), fresh pulp rate (g), moisture (g) or dry matter, so that results should be presented not only  relative dissimilarity.

Is possible to add (present those data?). Manuscript would be much more interesting for readers.

The discussion section presents a very good comparison of the obtained results with other results available in the data basis.

Similar remark to authors as in Introduction section with not-actual references:

(Venable and Lawlor 1980) or (Qaderi and Cavers 2002).

Could the authors can find more actual and replaced?

The obtained conclusions are clear and in accordance with results.

General opinion: after correction of results section  I think, that presented manuscript is a very valuable and should be published in Plants journal.

Author Response

Response to Reviewer 3 Comments

Point 1: I have only one question to this part of manuscript: is really need to use a little bit old references as: (Ratter 1992), (Delgado et al. 2007), or (Luzuriaga et al. 2006)? In my opinion not older than 10-years old should be used.

The collected experimental material and used methods do not raise any objections.

In my opinion the obtained results should be presented in tables as measured results with their units. If authors examined for example fresh fruit rate (g), fresh pulp rate (g), moisture (g) or dry matter, so that results should be presented not only  relative dissimilarity.

Is possible to add (present those data?). Manuscript would be much more interesting for readers.

The discussion section presents a very good comparison of the obtained results with other results available in the data basis.

Similar remark to authors as in Introduction section with not-actual references:

(Venable and Lawlor 1980) or (Qaderi and Cavers 2002).

Could the authors can find more actual and replaced?

The obtained conclusions are clear and in accordance with results.

General opinion: after correction of results section  I think, that presented manuscript is a very valuable and should be published in Plants journal.

Response: We changed.

We change many factors in Discussion, and put more recent reff.

We check all the references

And we think these old references are important too

we attached here the text with all the changs.

Round 2

Reviewer 1 Report

Thanks for working  on my comments 

Reviewer 3 Report

Authors of presented for review manuscript: “The influence of environmental features on the morphometric variation in Mauritia flexuosa L.f. fruits and seeds” changed everything I indicated in the review. Therefore, I find the work interesting and should be published in presented form.